# Impact of Comorbidity on the Duration from Symptom Onset to Death in Patients with Coronavirus Disease 2019: A Retrospective Study of 104,753 Cases in Pakistan

**DOI:** 10.3390/diseases11040176

**Published:** 2023-12-01

**Authors:** Haoqi Zhou, Jingyuan Wang, Naseem Asghar, Baosheng Liang, Qianqian Song, Xiaohua Zhou

**Affiliations:** 1Department of Biostatistics, School of Public Health, Peking University, Beijing 100191, China; 2211110200@stu.pku.edu.cn (H.Z.); wjy123@bjmu.edu.cn (J.W.); songqq0201@stu.pku.edu.cn (Q.S.); azhou@math.pku.edu.cn (X.Z.); 2Department of Statistics, Abdul Wali Khan University, Mardan 23200, Pakistan; nsmsghr@gmail.com; 3Chongqing Research Institute of Big Data, Peking University, Chongqing 401121, China

**Keywords:** COVID-19, comorbidities, hypertension, diabetes, chronic lung disease

## Abstract

(1) Background: The evidence indicates that comorbidities are associated with an increase in the risk of death from coronavirus disease 2019 (COVID-19). It is unclear whether such an association is different for various combinations of chronic disease comorbidities. (2) Methods: From 16 March 2020 to 30 November 2021, 104,753 patients with confirmed COVID-19 from Khyber Pakhtunkhwa Province, Pakistan, were studied to determine the association between comorbidities and the duration from symptom onset to death in patients with COVID-19 by stratifying their comorbidity status. (3) Results: The patients with comorbidities had an 84% (OR, 0.16; 95% CI, 0.14 to 0.17) decrease in the duration from symptom onset to death, as opposed to patients without a comorbidity. Among the patients with only one comorbidity, chronic lung disease (OR, 0.06; 95% CI, 0.03 to 0.09) had a greater impact on the duration from symptom onset to death than hypertension (OR, 0.15; 95% CI, 0.13 to 0.18) or diabetes (OR, 0.15; 95% CI, 0.12 to 0.18). The patients with both hypertension and diabetes had the shortest duration (OR, 0.17; 95% CI, 0.14 to 0.20) among the patients with two comorbidities. (4) Conclusions: Comorbidity yielded significant adverse impacts on the duration from symptom onset to death in COVID-19 patients in Pakistan. The impact varied with different combinations of chronic disease comorbidities in terms of the number and type of comorbidities.

## 1. Introduction

COVID-19 is caused by SARS-CoV-2, which is an infectious viral disease [1]. The virus has caused a worldwide pandemic and seriously threatened public health since it was first reported in December 2019. As of 16 August 2023, over 769.7 million confirmed cases of COVID-19 and 6.9 million deaths have been reported globally [2].

Based on the previous literature, approximately 20–60.9% of confirmed COVID-19 patients reported at least one comorbidity and more than one-third of patients had multiple comorbidities [3,4]. Patients with comorbidities are often in poor health, such as the elderly and people who smoke. Therefore, in order to protect the patients at high risk, the analysis of comorbidities in COVID-19 patients is of great importance. The influence of comorbidities on COVID-19 outcomes has been recognized since the earliest days of the pandemic [5,6]. It has been discovered in many studies that comorbidities are associated with a poor prognosis in COVID-19 patients. For example, a systematic review and meta-analysis conducted in 2020 [6] included 25 studies from December 2019 to July 2020, where data regarding 11 pre-existing comorbidities and 65,484 patients with COVID-19 from North America, Europe, Asia, and Africa were analyzed. The results showed that, compared with patients without comorbidities, those with cardiovascular disease, hypertension, diabetes, congestive heart failure, chronic kidney disease, and cancer were associated with a significantly greater risk of mortality from COVID-19, with the risk ratios ranging from 1.47 to 3.25. Among the various comorbidities, hypertension and diabetes were the most prevalent, with an increased risk of 82% and 48%, respectively, resulting in a substantial influence on the patients’ health status. Chronic obstructive pulmonary disease, which is directly associated with COVID-19, especially in patients with pneumonia, was also highly associated with the risk of severe disease. Diana et al. [7] investigated the association between chronic respiratory diseases and severe COVID-19 outcomes in a systematic review and meta-analysis. The study analyzed 22 studies, including 13,184 COVID-19 patients from multiple countries, and found that respiratory diseases (especially chronic obstructive pulmonary disease, OR = 4.21) and smoking (OR = 1.98) were significantly associated with severe COVID-19 outcomes.

Since the World Health Organization declared that COVID-19 is no longer a global health emergency, COVID-19 itself has received less attention than in past years. But a new topic, post-COVID-19 condition (PCC), is receiving more and more attention in the post-COVID-19 era. PCC is a complex heterogeneous disorder that has affected the lives of millions of people globally, and an estimated 10–15% of those infected with SARS-CoV-2 may develop post-COVID-19 condition, which impacts the survivors’ quality of life [8]. Unfortunately, chronic comorbidities are also closely associated with PCC according to many studies. For example, a systematic review and meta-analysis published in 2023 [9] evaluated the association between demographic characteristics and comorbidities and PCC using data from December 2022 to February 2023, the period in which most countries in the world experienced large-scale infection by SARS-CoV-2. The results from 860,783 patients demonstrated the high association between chronic obstructive pulmonary disease (COPD), diabetes, and PCC, where the odds ratios were 1.38 and 1.06, respectively (all with a *p*-value < 0.05). However, it is not clear whether such an association is different with different combinations of multiple comorbidities. All of these studies highlight the importance of paying attention to populations with specific comorbidities and tailoring treatments to meet their needs.

Compared with most developed nations, which were equipped with state-of-the-art healthcare systems and facilities for testing, the severity of COVID-19 was worse in most low-income areas because deprived populations often have scarce access to health services, even in normal circumstances [10]. As a low-income country, Pakistan faced a rise in chronic conditions due to the rapid aging of the population, putting it at increased risk of severe COVID-19. Since the first confirmed case was reported in Pakistan in January 2020, there have been 1.5 million confirmed cases of COVID-19, with thirty thousand deaths [2]. As for the association between comorbidities and the outcomes in COVID-19 patients in Pakistan, most analyses were just descriptive studies, and the investigation into this topic was not deep enough. For example, Song et al. [11] studied the effect of vaccination on the duration from symptom onset to death and preliminarily discussed the association between comorbidities and SOD. The results showed that the odds ratios for comorbidities ranged from 0.53 to 0.73 (all *p*-values < 0.05), although the differences between the number and type of comorbidities were not investigated. Therefore, a better understanding of the association between comorbidities and the SOD could facilitate clinical decision-making.

The main purpose of this study was to investigate the impact of comorbidities on the length of time from symptom onset to death in patients with COVID-19 according to the number and type of the comorbidities, which could help in the management of COVID-19 in locations with limited medical resources.

## 2. Materials and Methods

### 2.1. Data Sources and Data Extraction

The data used in this investigation were acquired from the office of the Directorate General Health Services, KP, Pakistan, which is responsible for the health system at all levels, based on the primary healthcare approach through the district health system, and aims to ensure universal access to quality healthcare in KP Province [11]. Briefly, the dataset included all hospital attendance records reported from all districts of KP Province to the Directorate during the four waves of SARS-CoV-2 infection. This study was approved by the office of the Additional Director General Health, Khyber Pakhtunkhwa. The diagnosis of COVID-19 was based on the results of real-time reverse transcriptase polymerase chain reaction (PCR) assays from nasal and pharyngeal swabs [12]. A total of 178,222 confirmed cases were collected; we excluded non-hospital data (68,605), data with logical errors (e.g., the date of symptom onset was later than the date of death, 268), and missing data (4396). Finally, 104,753 cases with complete data were extracted from 16 March 2020 to 30 November 2021. Figure 1 shows a flow chart of the data selection process. We also roughly divided the data into four waves, as well as the periods of Alpha and Delta infection, based on official media reports [13] and date of symptom onset. According to official media reports and the trends in weekly new cases, the range of the first wave was defined as being from 16 March 2020 to early August 2020; the second wave was from early November 2020 to mid-February 2021; the third wave started in March 2021 and ended in June 2021; and the fourth wave was from early July 2021 to late October 2021. The periods of Alpha and Delta infection in KP were roughly from 26 February 2021 to 15 May 2021 and 7 July 2020 to 30 November 2021, respectively.

Up until the end of data collection, the status of the patients was recorded, including active, recovered, and dead patients. Active means that the patients were still hospitalized at the last observation. We chose the duration from symptom onset to death (SOD) as the outcome of interest. The comorbidities included hypertension, diabetes, and chronic lung disease, which were determined based on patients’ self-reports on admission. We initially treated comorbidities as a categorical variable, and we subsequently grouped them according to the number of comorbidities. We also listed all of the combinations of different comorbidities. The date of symptom onset was defined as the date when COVID-19-related symptoms first occurred. The case fatality rate was defined as the proportion of death among all the confirmed cases over a certain period. The primary outcome was death, patients with the status of recovered or active at the last follow-up were treated as censored, and the survival observation of the outcome was defined as the duration from symptom onset to death.

### 2.2. Statistical Analyses

Continuous variables were presented as the mean ± standard deviation (SD), and categorical variables were described with the frequencies (percentages) according to the category of the outcome. The t-test and Chi-square test were used to compare continuous and categorical variables between the different groups, respectively. We applied the Kaplan–Meier method and the log-rank test to estimate and compare the survival probability between patients with and without comorbidity. Accelerated failure time (AFT) models [14] were applied to estimate the association between comorbidity and the SOD. We did not choose to use the Cox proportional hazards model because the global Schoenfeld test on the proportional hazard assumption was rejected for the data. The odds ratio (OR) and 95% confidence interval were applied to quantify the impact. Four parametric AFT models, including Weibull, Gamma, log-normal, and log-logistic, were fitted. The Akaike information criterion (AIC) was adopted for the model selection. Age, gender, and symptoms were adjusted in the final AFT model because they had been recognized as the risk factors for death in COVID-19 patients. Moreover, we conducted stratified analyses by age, gender, wave, and type of virus. 

As an alternative analysis strategy, logistic regression and propensity score matching (PSM) were further conducted to compare with and validate the obtained results. The PSM by gender, age group, symptom, and type of virus at a 1:2 ratio was employed to balance the distribution difference between the patients with and without comorbidity. All the analyses were performed with R (version 4.2), and a two-sided level of 0.05 was considered to be statistically significant.

## 3. Results

### 3.1. Characteristics of Study Participants by Death

A total of 104,753 patients with confirmed COVID-19 were included in the present study, and of these, 5600 died and 99,153 survived. Of these cases, 64,667 (61.7%) patients were male, and the mean (SD) age was 42.7 ± 18.0 years. In total, 30,078 (28.7%) cases had COVID-19-related symptoms, with the most common symptom being fever (25.1%), followed by cough (21.4%) and respiratory issues (11.5%), while sore throat (9.7%), headache (3.7%), and diarrhea (1.5%) were less common. There were also many differences between the dead patients and those who survived. Compared with the cases of survival, the patients who died had significantly higher age, were more likely to be female, and were more likely to report all the COVID-19-related symptoms, including fever, sore throat, cough, diarrhea, respiratory issues, and headache (*p <* 0.001, Table 1). The patients who died also had a higher probability of comorbidities (*p <* 0.001).

Overall, 4604 (4.4%) patients reported having at least one comorbidity. Hypertension (3.4%) was the most commonly reported comorbidity, and the prevalence of diabetes and chronic lung disease was 2.6% and 0.5%, respectively. At least one comorbidity was seen more commonly in the patients who died than in those who survived (18.5% vs. 3.0%, *p <* 0.001). We also found that the patients who died were more likely to have hypertension (21.9% vs. 2.3%), followed by diabetes (15.7% vs. 1.9%) and chronic lung disease (2.3% vs. 0.4%).

### 3.2. Characteristics of Study Participants by Comorbidities

As shown in Table 2, comorbidities were more likely to be found among male patients than female patients, both for one (61.9% vs. 38.1%) and two or more (53.1% vs. 46.9%) comorbidities. The proportion of people with comorbidities increased with age, with more than half of those over 60 having comorbidities. We observed a higher proportion of patients with comorbidities in the first (38.5%) and third waves (29.4%) than in the second and fourth waves. Moreover, patients infected with the Alpha strain had a higher prevalence of one comorbidity (24.2% vs. 13.1%), while two or more comorbidities were more common in those infected with the Delta strain (21.2% vs. 27.3%). The length of the SOD in patients with one comorbidity was similar to that in patients with two or more comorbidities (13.9 days vs. 13.7 days), and the difference was insignificant. There were no significant differences in the prevalence of hypertension, diabetes, or chronic lung disease between males and females. The older the age, the higher the prevalence of hypertension. Diabetes (53.1%) and chronic lung disease (59.5%) were the most common comorbidities for patients between the ages of 18 and 59. The length of the SOD was shorter in patients with hypertension and diabetes than in patients without comorbidity. More detailed information about the patients’ characteristics and comorbidities is presented in Table 2.

### 3.3. Impact of Comorbidity on Length of the SOD

As shown in Figure 2, patients with comorbidity had a lower survival probability than patients without comorbidity (*p* < 0.0001). We divided the patients according to the number of comorbidities and found that the survival probability of patients with two comorbidities was significantly lower than that of patients without or with three comorbidities (*p* < 0.0001). In addition, patients with three comorbidities had a significantly lower survival probability than patients without comorbidities. Although patients with two comorbidities had a lower survival probability than patients with a single comorbidity, the difference was minor. However, since the size of the group with three comorbidities was relatively small at 318 (proportion 0.3%), we think that the Kaplan–Meier (K–M) curve of the group with three comorbidities is not comparable with the other K–M curves. 

Table 3 presents the relationship between comorbidity and the length of the SOD. The patients with comorbidity had an 84% (OR, 0.16; 95% CI, 0.14 to 0.17) decrease in the SOD, as opposed to patients without comorbidity. Compared with patients without comorbidity, the length of the SOD in patients with one type of comorbidity and that in patients with two or more types of comorbidities decreased by 86% (OR, 0.14; 95% CI, 0.12 to 0.16) and 82% (OR, 0.18; 95% CI, 0.16 to 0.21), respectively. We also investigated the impact of different types of comorbidities on the SOD. Among the patients with a single comorbidity, cases with chronic lung disease (OR, 0.06; 95% CI, 0.03 to 0.09) had a greater impact on the length of the SOD than those with hypertension (OR, 0.15; 95% CI, 0.13 to 0.18) or diabetes (OR, 0.15; 95% CI, 0.12 to 0.18). Patients with both hypertension and diabetes had the shortest length of the SOD (OR, 0.17; 95% CI, 0.14 to 0.20), followed by patients with both hypertension and chronic lung disease (OR, 0.19; 95% CI, 0.10 to 0.34) and diabetes and chronic lung disease (OR, 0.33; 95% CI, 0.10 to 1.09). Patients with three comorbidities exhibited a 47% decrease in the SOD, although the impact was insignificant (OR, 0.53; 95% CI, 0.26 to 1.08).

### 3.4. Stratified Analyses

The stratified analyses are illustrated in Figure 3. The association between comorbidity and length of the SOD did not change substantially according to gender, age, the wave of COVID-19, or the type of virus. In each subgroup stratified by gender, age, wave of COVID-19, or type of virus, the comorbidity was significantly associated with a shorter SOD (*p <* 0.05). We observed that the impact of comorbidity on the SOD was greater in the early period of the epidemic (OR, 0.01; 95% CI, 0 to 0.01), and it then decreased over time (OR, 0.72; 95% CI, 0.60 to 0.86). The impact of comorbidity was higher during the Alpha period (OR, 0.38; 95% CI, 0.31 to 0.45) than during the Delta period (OR, 0.73; 95% CI, 0.60 to 0.87). A greater impact of comorbidity was also observed among younger patients (OR, 0.03; 95% CI, 0.02 to 0.03) than older patients (OR, 0.34; 95% CI, 0.30 to 0.38). 

### 3.5. Sensitivity Analyses

To evaluate the sensitivity of the potential conclusion concerning the selection of survival methods, we also conducted additional analyses using the logistic regression and propensity score matching methods to validate and compare with the previous results. Table 4 presents the results of the logistic regression in which we treated the patients’ cure status as a binary outcome and their gender, age group, type of symptom, and type of virus were adjusted in the logistic model. Compared with the patients without comorbidity, the risk of death significantly increased by 3.8 times in the patients with one comorbidity (OR, 4.8; 95% CI, 4.5 to 5.3) and 2.9 times in the patients with two or more comorbidities (OR, 3.9; 95% CI, 3.5 to 4.3). Table 5 shows the results of the PSM. For the impact of comorbidity on the length of the SOD, the general conclusions were consistent with those before the PSM. The OR was 0.15 (95% CI, 0.14 to 0.17) for patients with comorbidity, 0.15 (95% CI, 0.12 to 0.16) for patients with one comorbidity, and 0.16 (95% CI, 0.13 to 0.19) for patients with two or more comorbidities.

## 4. Discussion

This study systematically evaluated the impacts of comorbidities and different types of comorbidities on the SOD among confirmed COVID-19 patients in KP. In the present study, hypertension was the most common comorbidity, followed by diabetes and chronic lung disease. Compared with patients without comorbidity, patients with one or more comorbidities had a shorter SOD.

Although there was variation in terms of the population and region, this study further provided evidence that hypertension was the most prevalent comorbidity among COVID-19 patients and that the prevalence of pre-existing respiratory disease was relatively low, as previous studies have reported [15,16,17]. Hypertension and diabetes have been recognized as risk factors for unfavorable progression in COVID-19, and patients with hypertension and diabetes exhibited a 42% and 54% increase in the relative risk of death from COVID-19, respectively [18]. A possible hypothesis for the pathophysiological mechanism related to the higher risk of death among patients with pre-existing chronic diseases is the increased allostatic load. Chronic conditions usually cause the dysregulation of major physiological systems, including the hypothalamic–pituitary–adrenal axis, the sympathetic nervous system, and the immune system [19]. The chronic nature of these systems induces “wear and tear” in the body’s regulatory system [20], which leads to the accumulation of pro-inflammatory cytokines and affects the cellular immune system. As a result, individuals with chronic comorbidities become susceptible to severe complications of SARS-CoV-2 and to death. It is also worth noting that such an association with this type of virus is not new. For example, seasonal influenza, SARS-CoV, and Middle Eastern Respiratory Syndrome-CoV are also associated with an increased death risk in patients with pre-existing chronic comorbidities [21,22].

The high association between hypertension/diabetes and the death risk could also be explained by the role of angiotensin-converting enzyme 2 (ACE2) in the process of SARS-CoV-2 invading cells. The entry of SARS-CoV-2 into cells is mediated via the binding of viral spike glycoproteins to cellular ACE2 [23]. Angiotensin-converting enzyme inhibitors and angiotensin receptor blockers are commonly used in hypertension patients, which may promote the expression of ACE2 [24,25], resulting in a worse prognosis for COVID-19. Among diabetes patients, the ACE2 expression is also increased, and hyperglycemia can induce the glycation of ACE2, which may also increase the entry of SARS-CoV-2 into the cells, leading to increased inflammation and hyper-immune responses [26,27]. Therefore, the chances of contracting the disease are increased. Moreover, previous studies showed that a high expression of ACE2 in specific organs was associated with organ failure in COVID-19 patients [28,29].

All of these hypotheses might help explain the greater propensity of patients with hypertension and diabetes to develop more severe disease among confirmed COVID-19 patients. Renin–angiotensin–aldosterone system (RAAS) blockers, which are used widely in most patients with pre-existing hypertension and diabetes, have also been postulated to increase the risk of developing more severe and fatal COVID-19 disease [30]. In this study, chronic lung disease had the greatest impact on the SOD. This is not surprising, because chronic lung disease usually results in airway inflammation and remodeling, variable alveolar destruction, and emphysema, which decrease the sensitivity of lung tissues to SARS-CoV-2 [31]. But the association between pre-existing respiratory disease and the risk of death among patients with COVID-19 was inconsistent with previous studies. A large representative community cohort reported that COVID-19 patients with COPD had a 54% increase in the risk of death, although there was no evidence that asthma was associated with an increased risk of death [32]. Another representative population cohort, the OpenSAFELY study, found that patients with asthma had a 13% increased risk of death under the adjustment using age and gender, while the association was insignificant under a fully adjusted model [33].

Studies indicate that chronic diseases usually coexist, and about 12.0% of people have two or more chronic conditions in Pakistan [34]. Compared with single morbidity in isolation, individuals with more than one chronic condition have poorer health. In our study, patients with two comorbidities had a shorter SOD compared with those with no or only one comorbidity, which was consistent with findings from China [35]. There are also studies revealing that late-stage chronic lung diseases were associated with a high prevalence of comorbidities, such as cardiovascular disease and diabetes, which was associated with a poor prognosis after contracting COVID-19 [3,4]. However, we observed that one or two comorbidities seemed to have a greater impact on the SOD than three comorbidities, which might result from the relatively small sample size of patients with all three comorbidities. According to previous studies [36], being male and of older age were independently associated with a higher risk of death, consistent with the results found in this study. We also found that the impact of comorbidities on the SOD was greater in the early stages of the epidemic and decreased over time. One possible reason for this is that the COVID-19 vaccines were available, and the study indicated that the vaccines provided excellent protection against severe COVID-19 [37]. It is known that there was a higher hospital admission or emergency care attendance risk for cases infected by the Delta variant compared with the Alpha variant [38], but in this study, among patients with the Delta variant, the impact of comorbidities on the SOD was lower than that of the Alpha variant. So far, few studies have evaluated the association between comorbidities and the SOD in terms of different COVID-19 virus variants, and more research is needed to confirm the findings in the future.

We acknowledge some specific limitations in this study. Firstly, the information concerning comorbidities was self-reported, and recall bias can inevitably result in the underestimation of comorbidities; moreover, the severity of the comorbidities was not classified. Secondly, due to the lack of detailed records or information about blood pressure and glucose, this study failed to evaluate the impact of blood pressure and glucose on the length of the SOD. Thirdly, we did not further classify chronic lung disease into different types of lung diseases due to the lack of relevant information. Fourthly, the classification of the Alpha and Delta variants was based on the reports from the epidemic period rather than the sequencing results, because the sequencing data were unavailable. Finally, although the evidence from this retrospective study was not as strong as that from prospective studies, it could provide a reference for clinical decision-making. There are also many other aspects to be explored in future analyses. For example, apart from the SOD, the time between the symptoms of disease onset and recovery or ICU admission is also an important indicator that reflects the severity of COVID-19 that can be further analyzed to evaluate the impact of comorbidity. The vaccination status of patients is also available in our dataset, making it possible to analyze the impact of comorbidity on vaccination effectiveness. In addition, since COVID-19 is no longer a global health emergency, PCC is also an important issue for most patients who became infected in the post-COVID-19 era. Therefore, in future analyses, PCC data could be collected and the association between comorbidities and PCC could be further analyzed for the better management of patients experiencing PCC.

## 5. Conclusions

Comorbidities had a significant adverse impact on the length of the SOD among confirmed COVID-19 patients. Chronic lung disease had a greater impact than hypertension and diabetes. The impact also varied with the number or combination of comorbidities. The length of the SOD in patients with two comorbidities was significantly shorter than in those with one or three. Among the patients with two comorbidities, those with hypertension and diabetes had the shortest SOD. These findings can help identify those patients who may be more likely to develop serious adverse outcomes and promote the rational allocation of limited medical resources.

## Figures and Tables

**Figure 1 diseases-11-00176-f001:**
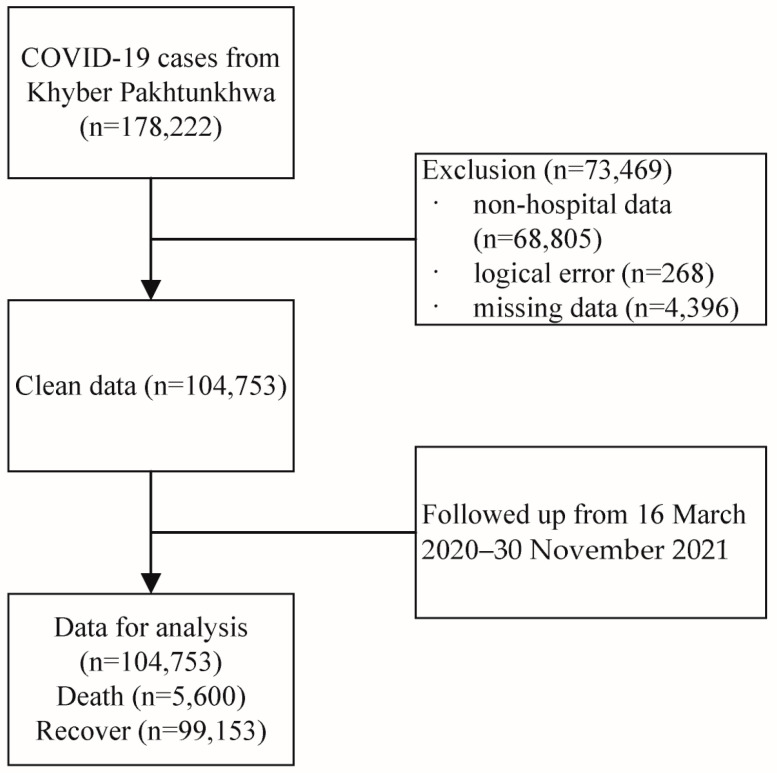
Flowchart of the data selection process.

**Figure 2 diseases-11-00176-f002:**
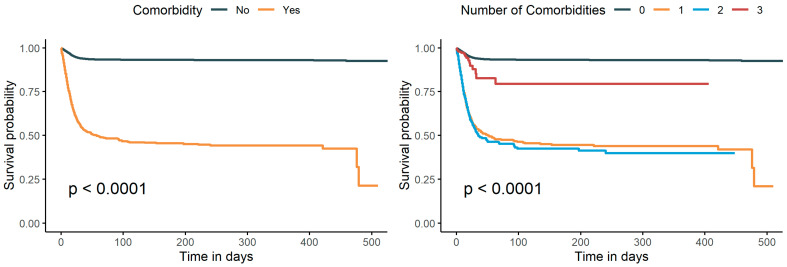
Survival curves of the SOD under different combinations of comorbidities.

**Figure 3 diseases-11-00176-f003:**
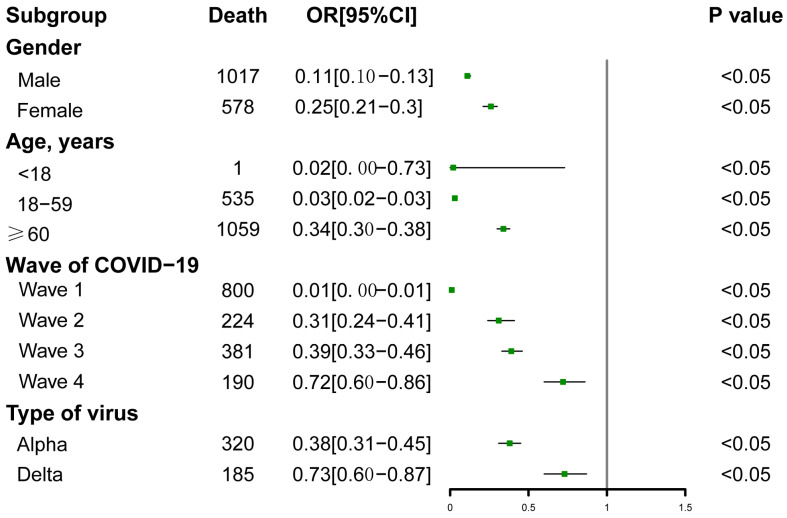
Stratified analyses of the associations between comorbidity and the SOD among COVID-19 patients.

**Table 1 diseases-11-00176-t001:** Characteristics of the study participants by death.

Characteristic	Overall, *n* (%)	Death, *n* (%)		*p*-Value
		No	Yes	
N	104,753	99,153	5600	
Age (year)	42.7 ± 18.0	41.6 ± 17.6	62.0 ± 13.7	<0.001
Gender				<0.001
Male	64,667 (61.7)	61,415 (61.9)	3252 (58.1)	
Female	40,086 (38.3)	37,738 (38.1)	2348 (41.9)	
Symptoms				<0.001
No	74,675 (71.3)	73,046 (73.7)	1629 (29.1)	
Yes	30,078 (28.7)	26,107 (26.3)	3971 (70.9)	
Type of symptom				
Fever	26,268 (25.1)	22,877 (23.1)	3391 (60.6)	<0.001
Sore throat	10,207 (9.7)	9304 (9.4)	903 (16.1)	<0.001
Cough	22,435 (21.4)	19,198 (19.4)	3237 (57.8)	<0.001
Diarrhea	1521 (1.5)	1365 (1.4)	156 (2.8)	<0.001
Respiratory issues	12,073 (11.5)	8787 (8.9)	3286 (58.7)	<0.001
Headache	3889 (3.7)	3474 (3.5)	415 (7.4)	<0.001
Comorbidity				<0.001
No	100,149 (95.6)	96,144 (97.0)	4005 (71.5)	
Yes	4604 (4.4)	3009 (3.0)	1595 (28.5)	
Type of comorbidity				
Hypertension	3551 (3.4)	2325 (2.3)	1226 (21.9)	<0.001
Diabetes	2728 (2.6)	1850 (1.9)	878 (15.7)	<0.001
Chronic lung disease	574 (0.5)	446 (0.4)	128 (2.3)	<0.001
Number of comorbidities				<0.001
0	100,149 (95.6)	96,144 (97.0)	4005 (71.5)	
1	2673 (2.6)	1695 (1.7)	978 (17.5)	
2	1613 (1.5)	1016 (1.0)	597 (10.7)	
3	318 (0.3)	298 (0.3)	20 (0.4)	
Epidemic wave				<0.001
1st wave	22,372 (21.4)	21,219 (21.4)	1153 (20.6)	
2nd wave	16,236 (15.5)	15,462 (15.6)	774 (13.8)	
3rd wave	39,218 (37.4)	36,972 (37.3)	2246 (40.1)	
4th wave	26,927 (25.7)	25,500 (25.7)	1427 (25.5)	
Type of virus				<0.001
Alpha	32,325 (30.9)	30,470 (30.7)	1855 (33.1)	
Delta	26,719 (25.5)	25,306 (25.5)	1413 (25.2)	

**Table 2 diseases-11-00176-t002:** Characteristics of the study participants by comorbidity.

Characteristic	Number of Comorbidities		HTN Only	DM Only	CLD Only	HTN and DM	HTN and CLD	DM and CLD	HTN, DM, and CLD
	1	≥2							
Gender									
Male	1655 (61.9)	1025 (53.1) *	1031 (62.4)	536 (60.2)	88 (67.2)	762 (51.2) *	59 (62.8)	19 (61.3)	185 (58.2)
Female	1018 (38.1)	906 (46.9)	620 (37.6)	355 (39.8)	43 (32.8)	726 (48.8)	35 (37.2)	12 (38.7)	133 (41.8)
Age group									
<18	5 (0.2)	9 (0.5) *	2 (0.1) *	2 (0.2) *	1 (0.8) *	1 (0.1) *	0 (0.0) *	1 (3.2) *	7 (2.2) *
18–59	1223 (45.8)	823 (42.6)	672 (40.7)	473 (53.1)	78 (59.5)	562 (37.8)	24 (25.5)	11 (35.5)	226 (71.1)
≥60	1445 (54.1)	1099 (56.9)	977 (59.2)	416 (46.7)	52 (39.7)	925 (62.2)	70 (74.5)	19 (61.3)	85 (26.7)
Epidemic wave									
1st wave	1030 (38.5)	558 (28.9) *	615 (37.3) *	330 (37.0) *	85 (64.9) *	459 (30.8) *	54 (57.4) *	23 (74.2) *	22 (6.9) *
2nd wave	501 (18.7)	371 (19.2)	312 (18.9)	172 (19.3)	17 (13.0)	348 (23.4)	14 (14.9)	1 (3.2)	8 (2.5)
3rd wave	787 (29.4)	470 (24.3)	491 (29.7)	273 (30.6)	23 (17.6)	445 (29.9)	16 (17.0)	3 (9.7)	6 (1.9)
4th wave	355 (13.3)	532 (27.6)	233 (14.1)	116 (13.0)	6 (4.6)	236 (15.9)	10 (10.6)	4 (12.9)	282 (88.7)
Type of virus									
Alpha	648 (24.2)	410 (21.2) *	415 (25.1) *	215 (24.1) *	18 (13.7) *	389 (26.1) *	15 (16.0) *	3 (9.7) *	3 (0.9) *
Delta	349 (13.1)	528 (27.3)	228 (13.8)	115 (12.9)	6 (4.6)	232 (15.6)	10 (10.6)	4 (12.9)	282 (88.7)
Length of SOD, median (95% CI)	13.9 (13, 14.9)	13.7 (12.9, 14.6)	13.7 (12.8, 14.6) *	14.5 (13.6, 15.5) *	13.7 (12.8, 14.7)	13.7 (12.9, 14.5) *	14.0 (13.1, 15.0)	17.0 (15.9, 18.1)	13.8 (13.2, 14.4) *

Abbreviations: HTN—hypertension; DM—diabetes mellitus; CLD—chronic lung disease. * indicates *p* < 0.05.

**Table 3 diseases-11-00176-t003:** Association of comorbidity with death in COVID-19 patients.

	Death, *n*	Case Fatality Rate	OR (95% CI)
Comorbidity	1595	0.346	0.16 [0.14–0.17]
Number of comorbidities			
1	978	0.366	0.14 [0.12–0.16]
≥2	617	0.320	0.18 [0.16–0.21]
Type of comorbidity			
HTN only	618	0.374	0.15 [0.13–0.18]
DM only	300	0.337	0.15 [0.12–0.18]
CLD only	60	0.458	0.06 [0.03–0.09]
HTN and DM	549	0.369	0.17 [0.14–0.20]
HTN and CLD	39	0.415	0.19 [0.10–0.34]
DM and CLD	9	0.290	0.33 [0.10–1.09]
HTN, DM, and CLD	20	0.063	0.53 [0.26–1.08]

Note: OR—odds ratio; HTN—hypertension; DM—diabetes mellitus; CLD—chronic lung disease. Gender, age, and symptoms were adjusted.

**Table 4 diseases-11-00176-t004:** Association of comorbidity with death using logistic regression.

Characteristic	OR (95% CI)	*p*-Value
Comorbidity		
0	Reference	Reference
1	4.8 [4.4–5.3]	<0.001
≥2	3.9 [3.5–4.3]	<0.001

Note: OR—odds ratio; gender, age group, symptom, and type of virus were adjusted.

**Table 5 diseases-11-00176-t005:** Association of comorbidity with death in COVID-19 patients after the PSM.

	Death, *n*	Case Fatality Rate	OR (95% CI)
Comorbidity	1595	0.346	0.15 [0.14–0.17]
Number of comorbidities			
1	978	0.366	0.13 [0.12–0.16]
≥2	617	0.320	0.16 [0.13–0.19]
Type of comorbidity			
HTN only	618	0.374	0.12 [0.10–0.14]
DM only	300	0.337	0.14 [0.11–0.18]
CLD only	60	0.458	0.08 [0.05–0.13]
HTN and DM	549	0.369	0.12 [0.10–0.15]
HTN and CLD	39	0.415	0.09 [0.05–0.16]
DM and CLD	9	0.290	0.20 [0.06–0.68]
HTN, DM, and CLD	20	0.063	2.17 [1.06–4.43]

Note: OR—odds ratio; HTN—hypertension; DM—diabetes mellitus; CLD—chronic lung disease.

## Data Availability

The data presented in this study are available on request from the corresponding author. The data are not publicly available due to privacy.

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
