# Peer review of "Impact of Comorbidity on the Duration from Symptom Onset to Death in Patients with Coronavirus Disease 2019: A Retrospective Study of 104,753 Cases in Pakistan"

_diseases, 2023, doi:10.3390/diseases11040176_

Round 1

Reviewer 1 Report (Previous Reviewer 1)

Comments and Suggestions for Authors

I have read the new submission and I think it is okay for publication in its present form.

Author Response

Please see the attached response letter, thank you

Reviewer 2 Report (New Reviewer)

Comments and Suggestions for Authors

Small requests for changes.  Review the capitalization thought. For example on line 160 remove the caps from Sore and Diarrhea.

Adjust and conform to the biographic format.  example, 272 versus 278 where some words have the reference immediately and others have a space.

Author Response

Please see the attached response letter, thanks.

Reviewer 3 Report (New Reviewer)

Comments and Suggestions for Authors

In the introduction the authors provide a very sound justification for undertaking this retrospective study. While several other cited studies have examined the influence of comorbidities in COVID-19 patients, and it is acknowledged that the presence of one or multiple comorbidities may significantly increase the risk of mortality, many of the studies have been carried out in countries where patients have good access to health care in a high socioeconomic environment. This study sought to investigate the impact of comorbidities on length of symptom onset to death in COVID patients living in a low-income country, Pakistan, where most previous analyses were merely descriptive.

The introduction is brief, but it does cite several of the relevant studies and provides an adequate background to the current study.

The study promised a very important practical benefit, to provide better management of COVID-19 in patients with comorbidities in a country with limited medical resources. 

Materials and methods: The dataset selection through four SARS-Cov-2 infection waves is outlined briefly and Fig 1 is very useful in providing a clear outline of the progress in selecting the large dataset used for analysis. The listing of all the combinations of comorbidities on patient outcome (symptom onset to death SOD) is useful and enabled by the analysis of a large dataset.

The statistical treatment of data followed appropriate standard analytical methods with sound reasons being provided for not employing the Cox proportional hazards model! Quantifying the impact using the odds ratio (OR) seems useful and appropriate. The reasons for choosing the Akaike information criterion (AIC) after examining four parametric AFT models is explained. The authors further performed rigorous analysis employing logistic regression and propensity score matching to validate the results obtained.

Results:

Table 1 is very informative and enables the reader to see that in patients that died the involvement of at least one comorbidity was involved with multiple comorbidities more often being associated with shorter SOD length. The presence of hypertension and diabetes as comorbidities often resulted in very poor outcomes for patients.

The survival corves of SOD under different combinations of comorbidities are starkly presented in Fig 2. 

In the stratified analyses presented in Fig 3 it is interesting that the impact of comorbidity on SOD was greater in the early period of the epidemic then decreased over time. There is clear evidence also that the impact of comorbidity was higher during the Alpha period than during the Delta period. Curiously, the impact of comorbidity was also observed to be greater in younger than in older patients.

The sensitivity analysis enabled a subtle view of the results confirming that compared with patients without comorbidity, risk of death significantly increased 3.8 times in patients with one comorbidity and 2.9 times in patients with more than two comorbidities!

The discussion is quite short, however it covers the  important  impact of the comorbidities examined in the study, in particular diabetes, hypertension and lung disease. The authors note that in the study, one or two comorbidities seemed to have greater impact on SOD than three comorbidities and that this unusual finding may have been related to the relatively small sample size of patients having three comorbidities! 

The discussion is quite detailed in parts and, where appropriate, regulation and dysregulation at the molecular level is considered for the systems involved.

The authors note some specific limitation in the study, one being that the information concerning comorbidities was from self -reporting where the recall bias was likely to result in underestimation of comorbidities and that severity was not classified.

The authors identify one other clear benefit from improved specific knowledge concerning the presence of comorbidities in potential patients who are likely to develop serious adverse outcomes because of their SARS-CoV-2 exposure and infection. With limited medical resources being available in Pakistan and indeed in similar countries where medical resources are limited, this knowledge should enable improved management of these medical resources.

Corrections and suggestions

L67 “…in the world have underwent the large-scale infection by SARS-CoV-2”. Need alter and suggesting …”….in the world have experienced large-scale infection by SARS-CoV-2,”

L 90 The principal purpose of this study was to investigate ,,,,

L 135 Chi-square test

L 160 breath issue – for breath issues  - maybe better to use “respiratory distress” or “respiratory issues” or  more broadly “respiratory problems” but note that “cough” is already used as a symptom?? Cough and breath issue(s) are used in the analysis so maybe just use an alternative name for breath issue(s)

L 329 Finally, the results from the retrospective study might not make important clinical or social significant, but it could provide reference for clinical decision making. This sentence needs to be restated because as it stands it is rather unclear! The future benefits for an improvement in clinical decision making are clear, however the first part of the sentence needs restating.

Perhaps include supplementary materials in the main text. Tables S1 and S2 do not add to the length of the paper and the readers then have direct access to the content that is mentioned in the paper under particular headings.

I suggest that this study has, indeed, provided a sound background to enable improved management of COVID-19 patients in Pakistan and it should be viewed with interest by health care workers in other similar low income countries where the COVID-19 pandemic experience may be similar.

Comments on the Quality of English Language

Generally the use of English expression is very good in this paper and I have indicated above, in comments to the authors, how specific improvements might be made.

Author Response

Please see the attached response letter, thanks.

This manuscript is a resubmission of an earlier submission. The following is a list of the peer review reports and author responses from that submission.

Round 1

Reviewer 1 Report

Comments and Suggestions for Authors

Impact of comorbidity on the duration from symptom onset to death in patients with coronavirus disease 2019.

The authors have explored the relationship between certain specific morbidities  and the period of onset to death in a Pakistani cohort. Specifically, they looked at Diabetes, hypertension, and Chronic Lung Disease, or combinations thereof. I found the paper to be of interest.

However, the authors should revisit the  logistic regressions they have presented in the results section, using one category as the baseline against which other Odds or risks  are calculated. Better still, they should compute relative risks

For instance, with reference to the effect of sex on, the authors are to compare Males relative to females, assigning unity to one of the categories. Where there are multiple levels  in the variable, one of the levels should be used as ‘unity’, and other ratios computed relative to that. For instance, when comparing the various waves with regards to morbidity, the risk ratios should be calculated relative to, say, the first wave, to which ‘unity’ will be ascribed.  The risks should be relative to the assigned baseline. This will make the table easier to interpret.

Otherwise, it is an interesting paper.

Author Response

Please see the attached response letter.  Thank you.

Reviewer 2 Report

Comments and Suggestions for Authors

The article presents a retrospective analysis of the medical conditions implicated in COVID-19 related mortality within a specific region in Pakistan, covering the period from March 2020 to November 2021. Based on the authors' findings, chronic lung disease poses the highest risk, followed by hypertension and diabetes mellitus. The data were collected during a time when no vaccines and treatments had yet been developed, and the mutations of COVID-19 were even more pronounced than at present. The potential contribution of this record towards combatting future COVID-19 outbreaks may be limited. Nevertheless, given the inevitability of pandemics recurring, it is crucial to make a record in the literature of the historical account of this outbreak caused by a solo virus, in a singular location, and at a specific moment for future reference. 

Author Response

Thank you for reviewing our manuscript. We appreciate your time and consideration. 

Reviewer 3 Report

Comments and Suggestions for Authors

This paper describes mainly time-to-death by different co-morbidity.

COVID-19 is an acute infectious disease, mortality arrives in a short time. The difference of time by subgroups of patients may be statistically significant but unlikely to make important clinical or social significant. Therefore the approach using time-to-event analysis is not appropriate.

Comments on the Quality of English Language

The abstract has a lot of unnecessary hyphens.

Author Response

(The authors gave the same response as above.)

Round 2

Reviewer 3 Report

Comments and Suggestions for Authors

Nil